# A Critical Comparison of Shape Sensing Algorithms: The Calibration Matrix Method versus iFEM

**DOI:** 10.3390/s24113562

**Published:** 2024-05-31

**Authors:** Cornelis de Mooij, Marcias Martinez

**Affiliations:** 1Faculty of Aerospace Engineering, Delft University of Technology, Kluyverweg 1, 2629 HS Delft, The Netherlands; 2Department of Mechanical and Aerospace Engineering, Clarkson University, 8 Clarkson Av., Potsdam, NY 13699, USA; mmartine@clarkson.edu

**Keywords:** calibration matrix, iFEM, shape sensing, load monitoring, strain sensors

## Abstract

Two shape-sensing algorithms, the calibration matrix (CM) method and the inverse Finite Element Method (iFEM), were compared on their ability to accurately reconstruct displacements, strains, and loads and on their computational efficiency. CM reconstructs deformation through a linear combination of known load cases using the sensor data measured for each of these known load cases and the sensor data measured for the actual load case. iFEM reconstructs deformation by minimizing a least-squares error functional based on the difference between the measured and numerical values for displacement and/or strain. In this study, CM is covered in detail to determine the applicability and practicality of the method. The CM results for several benchmark problems from the literature were compared to the iFEM results. In addition, a representative aerospace structure consisting of a twisted and tapered blade with a NACA 6412 cross-sectional profile was evaluated using quadratic hexahedral solid elements with reduced integration. Both methods assumed linear elastic material conditions and used discrete displacement sensors, strain sensors, or a combination of both to reconstruct the full displacement and strain fields. In our study, surface-mounted and distributed sensors throughout the volume of the structure were considered. This comparative study was performed to support the growing demand for load monitoring, specifically for applications where the sensor data is obtained from discrete and irregularly distributed points on the structure. In this study, the CM method was shown to achieve greater accuracy than iFEM. Averaged over all the load cases examined, the CM algorithm achieved average displacement and strain errors of less than 0.01%, whereas the iFEM algorithm had an average displacement error of 21% and an average strain error of 99%. In addition, CM also achieved equal or better computational efficiency than iFEM after initial set-up, with similar first solution times and faster repeat solution times by a factor of approximately 100, for hundreds to thousands of sensors.

## 1. Introduction

Modern structures face ever greater demands: they must be lighter, stronger, and last longer, while operators demand a reduction of downtime due to inspection and maintenance. One of the approaches under development for achieving these goals is the Digital Twin concept [1,2], which seeks to continuously monitor aircraft structures on an individualized basis. Sensor data is collected and fed into digital structural models to predict when and where damage may occur based on the historical use of the aircraft and established maintenance reports. Based on these predictions, maintenance and repairs can be carried out in a more targeted fashion, resulting in lower operational costs and less downtime.

Limited sensor data is one of the challenges this approach encounters. For instance, traditional strain gauges only measure at a single point and along a single axis. To measure in multiple directions, multiple strain gauges are collocated to form rosettes. Many “shape sensing” algorithms have been designed to translate limited data into a full picture of the (deformed) shape to provide structural engineers with the strain field required for life assessment of their critical assets [1,2,3].

Shape sensing was first mentioned in the scientific literature in 1983 by Carl G. Saunders [4], who used Moiré patterns and an interactive computer program to design better fitting prostheses [5,6,7]. A few years later, a similar method using laser speckle patterns was applied to structural problems with both experimental and numerical demonstrations by Weathers et al. [8]. In 1989, Maniatty et al. [9] identified the challenges of finding complete strain and displacement fields for entire structures, noting that small displacement errors lead to larger inaccuracies in stress and strain, as the displacements need to be differentiated. An alternative was to measure strains directly [10]; however, translating strain measurements into displacement distributions is an inverse problem, which has its own host of challenges. Inverse problems are a type of ill-posed problem whose solutions generally do not necessarily satisfy the conditions of uniqueness, existence, and stability [11,12,13,14].

After the initial illumination-based methods, which were unsuitable for larger scale applications, various alternative methods were proposed: minimization of an objective function with regularization [9,15,16], statistical methods based on Bayesian theory [15], fitting to a polynomial [16,17,18,19], dynamic programming [20], modal transformation [21], and direct integration of multicore optical fibers [22,23,24,25]. Each of these methods has its own drawbacks: most require manual tuning to determine parameters such as regularization constants, polynomial order, and/or the number of modes. The exception is direct integration of multicore optical fibers; however, it presents its own challenge: it only solves part of the problem, as it only determines the shape of the optical fiber that makes the measurements, not the shape of the entire structure to which it is attached [22,23,24,25].

Another approach considered by the scientific community is the inverse Finite Element Method (iFEM), initially introduced by Tessler et al. in 2001 for determining the structural response from in-flight strain measurements [26] and for Structural Health Monitoring (SHM) in 2002 [27,28]. They demonstrated it numerically on beam, plate, and shell structures in 2003 [29] and verified and documented it experimentally from 2003 to 2005 [30,31,32]. Two further studies showed how iFEM could use fiber optic strain gauges to detect structural anomalies [33] and for SHM applications [34]. Two more papers demonstrated iFEM for real-time SHM of spacecraft in 2007 and 2009 [35,36]. In 2019, de Mooij et al. performed a baseline study of iFEM for hexahedral elements based on the McNeal and Harder standard problem set, originally established for evaluating the accuracy of finite element programs [37].

Tessler started a collaboration on iFEM in 2011 with Italian researchers Gherlone, Cerracchio, Mattone, and Di Sciuva, first focusing on beam, truss, and frame elements [38,39], then exploring dynamic loads numerically [40]. They also studied the effects of sensor locations and the number of elements on accuracy [41]. In 2013, they discovered a severe limitation [42,43] with the earlier iFEM work that used the shell element iMIN3: this type of element could be very inaccurate, particularly for structures with large stiffness variations, such as sandwich structures and laminates, which are common in aerospace and space applications. They also presented a new element, iRZT, that can handle these stiffness variations based on the Refined Zigzag Theory [44,45].

Shell-based iFEM for shape sensing was introduced for maritime applications in 2016 by Kefal and Oterkus [46], who also introduced quadrilateral shell elements. They have since steadily produced additional numerical demonstrations on ships [47,48], a wing-shaped sandwich panel [49], and cylindrical marine structures [3]. Various other structures have been studied with iFEM by other researchers, including offshore drilling platforms [50] and wind turbines [51].

A method for shape sensing based on the calibration matrix algorithm (CM) [52] was described by Nakamura et al. in 2012 [53]. A mathematical relationship was established between nodal pressures, i.e., loads or forces, and the strains that would be measured at sensors distributed on a 2D plate, consisting of a rectangular matrix and a vector containing the nodal pressures. To solve for the nodal pressure, a generalized Moore-Penrose pseudo-inverse was determined to invert the relationship.

The CM algorithm presented in this paper builds upon Nakamura’s method in several ways: instead of solving directly for the large number of individual nodal pressures, the method solves for a relatively small number of basic load case coefficients, which reduces the risk of failure due to singular matrices when dealing with fine meshes. This also makes it possible to focus the solution on physically feasible solutions based on domain-specific knowledge about the structure and how it will deform. This method also removes the need for strain sensors to coincide with mesh nodes, allowing them to be placed arbitrarily instead. It also facilitates incomplete or unaligned measurement data; thus, measurements can be made in any direction and can include a subset of the strain components, which can be different for each sensor as long as the sensor distribution is the same for each load case. Lastly, the solution method for inverting the relationship between the measurements and the load case coefficients was simplified by replacing the computationally intensive Moore-Penrose inverse with the approximate normal equations’ solution for ordinary least squares [54].

## 2. Materials and Methods

This section shows the benchmark problems and the representative aerospace structure that were used to test the CM and iFEM algorithms and provides a summary of the results of both methods. This includes the assumptions that were made for each algorithm, their mathematical underpinnings, flowcharts showing the structure of each algorithm, a description of the convergence study that was carried out to evaluate the accuracy of the mesh, the method for establishing what constitutes a sufficient number of strain sensors, and the method for evaluating their relative computational efficiency.

Numerical demonstrations are presented to characterize and compare the accuracy and computational efficiency of each algorithm. These demonstrations are carried out using several benchmark problems from the literature [37,55], which are shown and described herein. Additionally, to investigate the limits of the CM and iFEM algorithms, an additional representative aerospace structure was analyzed.

Reconstructing the deformed structural shape based on a limited amount of strain data is an inverse problem, which makes it inherently impossible to perfectly reconstruct it without additional data or assumptions. Each algorithm solves this problem by making certain assumptions, which will be described, followed by the process and theoretical underpinnings of both algorithms.

The core of the CM algorithm is derived from the principle of superposition, and it is shown how the decomposition of an actual load case can be approximated based on limited sensor data. The iFEM algorithm is based on the minimization of an error function that compares the numerical displacements and strains to their numerical counterparts and smooths the displacements using Tikhonov regularization. It is also shown how the accuracy of the results was evaluated using the 2-norm error. Visual overviews of both algorithms are provided in the form of flowcharts.

The structures were discretized to create meshes, which are required for both algorithms. A 20-node hexahedral element with reduced Gaussian integration, each having 8 internal Gauss nodes, was implemented as part of an in-house developed code. Any discretization of a structure approximates its domain, so a mesh convergence study was performed to verify that the mesh for a representative aerospace structure was a good approximation of the structure, boundary conditions, and load cases that were used. For the benchmark problems, such a mesh convergence study was not needed, as the FEM results could be compared to the analytical solutions provided by MacNeal and Harder [55]. Additionally, a computational efficiency study was carried out for both methods to compare their relative performance.

### 2.1. Numerical Demonstration: Benchmark Problems

To show how each method can reconstruct structural deformation based on sensor data, both were demonstrated numerically for several benchmark problems from the literature [37,55]. These benchmark problems are illustrated in Figure 1 below:

The properties of the benchmark problems for shape sensing from de Mooij et al. [37] are reproduced in Table 1 below. Note that most of these problems have multiple variants, either by varying the element shape, boundary conditions, applied load or Poisson’s ratio:

### 2.2. Numerical Demonstration: Representative Aerospace Structure

Additionally, to investigate the limits of the CM and iFEM algorithms, a blade profile was used to analyze an additional structure that is representative of a more complex scenario. A twisted and tapered blade with a NACA 6412 cross-sectional profile, thickened uniformly by a factor of 5, is shown in Figure 2. The base has a length of 0.5 m, a width of 1.0 m, and a thickness of 0.5 m. The blade has a length of 5.0 m, a width that tapers from 1.0 m at the base to 0.5 m at the tip, and a twist that varies from 0 radians at the base to π/4 radians at the tip. The base and the blade are connected by a transition section with a length of 0.125 m. All sections use the same material properties: a Young’s modulus of 17.2 GPa and a Poisson’s ratio of 0.22. These properties were measured for a similar blade made of 45% glass fiber-reinforced polyphthalamide (PPA).

This structure was meshed with 648 hexahedral elements: 4 elements through the thickness (z axis), 6 elements along the width (y axis), and 27 elements along the length (x axis), which was divided into 4 elements along the length of the rectangular base, 22 elements along the length of the aerodynamic surface, and 1 element for the connecting section.

The surface of the base with the minimum x coordinate, at the right-hand side in Figure 2, is fully constrained. The load cases were applied to the lower surface of the aerodynamic section, i.e., the external surface of the blade with the lowest z coordinates, and were evaluated using linear FEM. The resulting nodal displacements were used to simulate the strain sensor data that were used as input for the CM algorithm. The illustration in Figure 3 shows the mesh of the structure and the locations of the simulated strain sensors, which were distributed randomly on the top and bottom surfaces (minimum and maximum 1% along the z-axis) of the blade:

### 2.3. Assumptions

In this study, the structures were assumed to be made of linear elastic Hookean materials, such that the deflection, angles, and strains are small enough to assume geometric linearity. In addition, it was assumed that the loads would either be constant or change slowly enough that any dynamic effects could be neglected.

The CM algorithm assumes that the deformation for the actual load case is a linear combination of the deformations resulting from a limited number of simpler load cases, referred to here as “basic load cases”. This assumption is possible because of the principle of superposition [56,57], which allows the total deformation to be represented as a linear combination of these simpler deformations when the load cases are linear and static, which was already assumed to be the case.

Deformation is caused by external forces and constraints, for which, in theory, a virtually infinite number of distributions are possible. However, in practice, the types of force distributions and constraints are often known and limited to a small number of options. For example, a particular aircraft wing will always be constrained in the same manner by the wing-box to which it is attached. The force distributions to which the wing will be exposed can be modeled with aerodynamic analysis software for the various flight conditions that the aircraft will encounter.

The iFEM algorithm minimizes an error function that compares numerical strains and/or displacements to their measured counterparts. It assumes that the solution is smooth, i.e., that the values at adjacent points in the structure are similar to each other. The degree of smoothness is controlled by including a regularization term in the error functional and assigning a greater or lesser weight to this term. In this study, Tikhonov regularization was used.

### 2.4. Calibration Matrix

The calibration matrix algorithm operates on the assumption that the actual load case is a linear combination of the various basic load cases. Thus, the resulting displacement distribution should be an equivalent linear combination (i.e., the same coefficients) of the displacement distributions of the basic load cases. Similarly, the resulting strain distribution should be an equivalent linear combination of the strain distributions of the basic load cases. The simulated strain measurements for the actual load case will also be an equivalent linear combination of the simulated strain measurement distributions for the basic load cases. This holds for any arbitrary collection of strain sensors. These strain sensors can also be placed in any location on the structure and in any orientation.

The equivalence between the displacement and strain combinations is due to the linear relationship between displacements and strains for small deformations [58]. The arbitrary locations on the structure are possible due to the linear relationship between the displacements of an arbitrary point within a finite element and the displacements of the nodes of that element [59]. The arbitrary sensor rotations are possible due to the linearity of the rotation.

When the basic load cases can be combined to reproduce the actual load case and the number of simulated sensor data points is greater than or equal to the number of basic load cases, then this method will be able to accurately reconstruct the actual load case [53].

Conventional FEM analyses were performed for each load case to find the resulting displacement and strain distributions. Assuming linear elastic behavior, the following equations describe how a linear combination, or superposition, of displacement distributions $u_{basic,k}$ from the basic load cases results in the actual displacement distribution $u_{actual}$, all measured in meters. Similarly, the basic strain distributions $E_{basic,k}$ can be combined to form the actual strain distribution $E_{actual}$, all dimensionless, using the same $c_{actual,k}$ coefficients because linear elastic deformation is assumed:(1)$$u_{actual}=\sum_{k=1}^{m}c_{actual,k}u_{basic,k}$$
(2)$$E_{actual}=\sum_{k=1}^{m}c_{actual,k}E_{basic,k}$$

In practice, it is impossible to measure all values in $u_{actual}$ and $E_{actual}$. This makes it impossible to invert Equations (1) and (2) to solve for $c_{actual}$, the collection of dimensionless $c_{actual,k}$ coefficients that describe how much each basic load case contributes to the actual load. Instead, the calibration matrix method determines a reconstructed coefficient vector $c_{reconstructed}$ that approximates the actual coefficients to produce a complete reconstruction of the deformation based on an incomplete set of (simulated) sensor data.

The simulated sensor data of the actual load case was gathered in a single sensor vector $s_{actual}$. This could, in principle, include both displacement (m) and strain (-) data, which can also be partial, i.e., missing certain displacement and strain components. For this study, only strain data was used.

While the choice of sensor data components and their order are not important, they should be consistent for all the load cases, including both the basic and actual load cases. This concept has a powerful side effect: if a sensor malfunctions during operation or between experiments, the analysis can continue by leaving out the missing or erroneous sensor data components for every load case. This makes the CM algorithm robust with respect to sensor failure.

The sensor vector is defined as follows:(3)$$s_{actual}={\left[\begin{array}{ccc} s_{1} & \cdots & s_{k} \end{array}\right]}^{T}$$

Here, k is the number of simulated sensor values. For the strain data used here, that means that $s_{1}$ through $s_{6}$ are the 6 strain components of strain sensor 1, $s_{7}$ through $s_{12}$ for strain sensor 2, etc. For each sensor, the strains that would have been measured for a particular load case are estimated using barycentric interpolation of the FEM strains from the four nearest Gauss points. To ensure realistic results for each sensor, the strains were only interpolated, not extrapolated. This was performed by clamping the barycentric weights to the range of 0 to 1, then renormalizing to ensure that the weights summed up to 1.

Completing the sensor simulations for all basic load cases results in a set of sensor vectors $s_{k}$. Similar to Equations (1) and (2), the actual sensor vector should equal a linear combination of these:(4)$$s_{actual}≅\sum_{k=1}^{n}c_{reconstructed,k}s_{k}$$

This relationship can be rewritten in matrix notation, as shown in Equation (5):(5)$$s_{actual}≅Sc_{reconstructed}=\left[\begin{array}{ccc} s_{1,1} & \cdots & s_{m,1}\\ \vdots & \ddots & \vdots \\ s_{1,n} & \cdots & s_{m,n} \end{array}\right]\left\{\begin{array}{c} c_{reconstructed,1}\\ \vdots \\ c_{reconstructed,m} \end{array}\right\}$$

Here, m is the number of basic load case coefficients and n is the number of sensor values. The sensor matrix S is in general, not square, so it is not invertible. It was assumed that the number of basic load case coefficients m is smaller than the number of sensor data points (per load case) n. The optimal approximate solution for an overdetermined system of equations can be found using ordinary least squares [60]:(6)$$c_{reconstructed}={\left(S^{T}S\right)}^{-1}S^{T}s_{actual}≅c_{actual}$$

The reconstructed dimensionless coefficients $c_{reconstructed}$ could match the actual coefficients $c_{actual}$ perfectly if the actual load could be represented exactly as a linear combination of the basic load cases.

In practice, there will be differences due to various reasons, such as numerical errors and insufficient sensor data. In real experiments or applications, additional errors may result from sensor placement and orientation inaccuracies, noise, etc. The relative error between the reconstructed and actual coefficients can be found with the following equation, which calculates a single 2-norm error value to evaluate the overall quality of the reconstruction. If the error is on the order of $10^{-2}$, i.e., 1%, it is accurate to approximately 2 decimal places. Smaller errors indicate a more accurate reconstruction:(7)$$\varepsilon_{coeff}=\frac{\sqrt{\sum_{k=1}^{m}{\left(c_{reconstructed,k}-c_{actual,k}\right)}^{2}}}{\sqrt{\sum_{k=1}^{m}{\left(c_{actual,k}\right)}^{2}\;}}$$

After determining the coefficients $c_{reconstructed}$, the vector of actual loads $f_{actual}$, measured in Newtons, can be reconstructed from the contributing basic load cases $f_{basic,k}$ using Equation (8).
(8)$$f_{reconstructed}=\sum_{i=k}^{n}c_{reconstructed,k}f_{basic,k}≅f_{actual}$$

For a perfect reconstruction, the reconstructed load $f_{reconstructed}$ will exactly equal $f_{actual}$. However, if the contributing load cases cannot be combined to exactly reproduce the actual load, there will be a difference. To evaluate the overall quality of the reconstruction, the relative error between the reconstructed and actual loads at each node i can be found with the 2-norm error as well, similar to Equation (7):(9)$$\varepsilon_{force}=\frac{\sqrt{\sum_{i=1}^{n}{\left(f_{reconstructed,i}-f_{actual,i}\right)}^{2}}}{\sqrt{\sum_{i=1}^{n}{\left(f_{actual,i}\right)}^{2}}}$$

Similarly, other reconstructed vectors can be compared to the corresponding actual vectors to determine the 2-norm errors for the displacements at the mesh nodes, the strains at the mesh Gauss nodes, and the simulated strains at the sensors. Substituting these vectors into Equation (9) results in alternative errors for $\varepsilon_{displacement}$, $\varepsilon_{strain}$ and $\varepsilon_{sensor\;strain}$.

The CM algorithm, as shown in Figure 4, was intended for structures with known geometry, material properties, and boundary conditions. This information is used to construct the coordinate and stiffness matrices for the FEM analyses. For both the basic load cases and the actual load cases, the corresponding load vectors are calculated, which are used together with the coordinate and stiffness matrices to carry out the FEM analyses. The resulting displacements are used to determine the strains at the Gauss points of the mesh, which are then used to simulate the data that would be measured by the distributed sensors.

The simulated sensor data from the basic and actual load cases is used to determine the basic load case coefficients for each actual load case, i.e., how much each basic load case appears to contribute to the actual load cases. Reconstructed versions of the actual load cases are made by multiplying each basic load case by its coefficient and adding up the results. Further FEM analyses are then run for the reconstructed load cases, the results of which are compared to the FEM results of the actual load cases to calculate the errors between the actual load distribution and the reconstructed load distribution.

The only unknowns were the actual load cases that were applied to and reconstructed for each benchmark problem and the representative aerospace structure. For each problem, the calibration matrix method needs to analyze an additional set of basic load cases to be able to reconstruct the applied loads. These basic load cases could be defined in various ways.

For the purposes of this study, the basic load cases were defined as follows: three basic load cases are defined for each external surface of the structural mesh that is not fully constrained, with the first basic load case of each such surface being loaded by a unit force aligned with the x-axis, the second by a unit force aligned with the y-axis, and the third by a unit force aligned with the z-axis. An example of such a basic load case is shown in Figure 5:

The simulated sensor data from these actual load cases were analyzed with the CM and iFEM algorithms, resulting in reconstructed distributions of deformation (displacements, strains, etc.).

### 2.5. iFEM Methodology

The iFEM methodology that was used for this study uses the same implementation as the study by de Mooij et al. from 2019 [37]. A summary of the methodology is provided here. A flowchart is shown in Figure 6 below as a graphical overview of the iFEM approach.

Different variants of iFEM optimize their estimates of the structural deformation by minimizing a least-squares error functional, which can be defined in various ways. The error function chosen for this study is shown in Equation (10). This error functional compares the numerical values for displacement, q (m), and/or strain, e (-), to the measured values, $q^{\varepsilon }$ and $e^{\varepsilon }$, and smooths the displacements using Tikhonov regularization. The Tikhonov regularization term that is used to enforce the smoothness of the solution is the $C_{0}{\left\|q\right\|}^{2}$ part of Equation (10). This error functional is minimized, resulting in a system of linear equations that can be solved for the numerical displacements, based on the sensor data. The coefficients (or weights), *C**q* ($m^{-2}$), *C**e* (-) and *C*_0_ ($m^{-2}$), can be used to minimize overall error with respect to different or multiple parameters [6,7,21,25].
(10)$$\Phi =C_{q}{\left\|q-q^{\varepsilon }\right\|}^{2}+C_{e}{\left\|e-e^{\varepsilon }\right\|}^{2}+C_{0}{\left\|q\right\|}^{2}\;\;\;\;\;\;\;\;\;\;\;\;\;\;\;\;\;\;\;=\frac{1}{V}\int C_{q}{\left[q-q^{\varepsilon }\right]}^{2}+C_{e}{\left[e-e^{\varepsilon }\right]}^{2}+C_{0}{\left[q\right]}^{2}dV$$

Exploring other variants of the iFEM error functional is outside of the scope of this study.

The numerical strains e (-) are related to the numerical displacements q (m) on a per-element basis by the linear strain-displacement matrix B ($m^{-1}$) as shown in Equation (11). It transforms the nodal displacements of an element into strains at a Gauss point within that element. $F_{x}^{y}$ (-) represents the derivative of the y component of the deformed position x (m) with respect to x (m). The notation $N_{x}^{1}$ ($m^{-1}$) indicates the derivative of the first shape function N (-) with respect to x. The lowercase n is the number of shape functions, which is equal to 20 for the 20 node hexahedral elements that were used in this study:(11)$$\begin{array}{c} e=Bq\\ ={\left[\begin{array}{c} \begin{array}{cc} \begin{array}{ccc} F_{x}^{x}N_{x}^{1} & F_{y}^{x}N_{y}^{1} & F_{z}^{x}N_{z}^{1}\\ F_{x}^{y}N_{x}^{1} & F_{y}^{y}N_{y}^{1} & F_{z}^{y}N_{z}^{1}\\ F_{x}^{z}N_{x}^{1} & F_{y}^{z}N_{y}^{1} & F_{z}^{z}N_{z}^{1} \end{array} & \begin{array}{ccc} F_{x}^{x}N_{y}^{1}+F_{y}^{x}N_{x}^{1} & F_{y}^{x}N_{z}^{1}+F_{z}^{x}N_{y}^{1} & F_{z}^{x}N_{x}^{1}+F_{x}^{x}N_{z}^{1}\\ F_{x}^{y}N_{y}^{1}+F_{y}^{y}N_{x}^{1} & F_{y}^{y}N_{z}^{1}+F_{z}^{y}N_{y}^{1} & F_{z}^{y}N_{x}^{1}+F_{x}^{y}N_{z}^{1}\\ F_{x}^{z}N_{y}^{1}+F_{y}^{z}N_{x}^{1} & F_{y}^{z}N_{z}^{1}+F_{z}^{z}N_{y}^{1} & F_{z}^{z}N_{x}^{1}+F_{x}^{z}N_{z}^{1} \end{array} \end{array}\\ \vdots \\ \begin{array}{cc} \begin{array}{ccc} F_{x}^{x}N_{x}^{n} & F_{y}^{x}N_{y}^{n} & F_{z}^{x}N_{z}^{n}\\ F_{x}^{y}N_{x}^{n} & F_{y}^{y}N_{y}^{n} & F_{z}^{y}N_{z}^{n}\\ F_{x}^{z}N_{x}^{n} & F_{y}^{z}N_{y}^{n} & F_{z}^{z}N_{z}^{n} \end{array} & \begin{array}{ccc} F_{x}^{x}N_{y}^{n}+F_{y}^{x}N_{x}^{n} & F_{y}^{x}N_{z}^{n}+F_{z}^{x}N_{y}^{n} & F_{z}^{x}N_{x}^{n}+F_{x}^{x}N_{z}^{n}\\ F_{x}^{y}N_{y}^{n}+F_{y}^{y}N_{x}^{n} & F_{y}^{y}N_{z}^{n}+F_{z}^{y}N_{y}^{n} & F_{z}^{y}N_{x}^{n}+F_{x}^{y}N_{z}^{n}\\ F_{x}^{z}N_{y}^{n}+F_{y}^{z}N_{x}^{n} & F_{y}^{z}N_{z}^{n}+F_{z}^{z}N_{y}^{n} & F_{z}^{z}N_{x}^{n}+F_{x}^{z}N_{z}^{n} \end{array} \end{array} \end{array}\right]}^{T}q \end{array}$$

The error functional is a sum of squares, so it is positive definite and can be minimized by differentiating and equating the result to zero. This is shown in detail by de Mooij et al. [37]. The resulting equations can be rewritten to obtain a linear system of equations, as shown in Equation (12):(12)$$\left[\int C_{q}I+C_{e}B^{T}B+C_{0}I\;dV\right]q=\left[\int C_{q}I\;dV\right]q^{\varepsilon }+\left[\int C_{e}B^{T}dV\right]e^{\varepsilon }$$

By defining matrix ***A*** (m) as the result of $\int C_{q}I+C_{e}B^{T}B+C_{0}I\;dV$ and vector ***b*** ($m^{2}$) as the result of $\left[\int C_{q}I\;dV\right]q^{\varepsilon }+\left[\int C_{e}B^{T}dV\right]e^{\varepsilon }$, Equation (12) can be written in short form as:(13)Aq=b

Because this system of equations is linear, it can be solved through Gaussian elimination.

### 2.6. FEM Methodology

Each load case was analyzed using the conventional Finite Element Method (FEM) to produce distributions of displacements and strains throughout the volume of the structure, which are represented by a collection of displacement components u, v and w along the x, y and z axes at each node of the structural mesh and a collection of strain components $e_{xx}$, $e_{yy}$, $e_{zz}$, $e_{xy}$, $e_{yz}$ and $e_{xz}$ at every Gauss point of the structure’s mesh, respectively.

A convergence study based on Richardson’s extrapolation [61,62] was carried out to verify the accuracy of the FEM analyses. Mesh quality influences the quality of any FEM results. The mesh elements should be sufficiently fine to be a good approximation of the real structure. Richardson’s extrapolation estimates the value the solution should converge towards as the mesh is refined and verifies that the analysis has converged.

#### Converge Study

The accuracy of FEM results is affected by the quality of the mesh. The elements should be sufficiently fine to approximate the real structure well. To assess this, a convergence study was carried out for the representative aerospace structure using Richardson’s extrapolation [61,62]. This technique finds an estimate for zero grid spacing (the value that the solution should asymptotically approach as the mesh is refined), the error band of this estimate, and whether the solution is in the asymptotic range of convergence. It does this by using three levels of mesh refinement, as presented in Table 2:

### 2.7. Sufficient Number of Strain Sensors

The methods that were analyzed for this study can use an arbitrary number of strain sensors. As such, several CM analyses were carried out to provide an indication of the number of strain sensors that were needed to obtain good results. These analyses were performed for 18 different load cases that were applied to the representative aerospace structure, which are listed in Table 3. As the precise criteria for sufficiently good results depend on the application, this can only be an indication. It is hypothesized that most of the errors, namely the coefficient, force, displacement, and strain errors, will rapidly decrease towards a minimum asymptote as the number of strain sensors increases.

The sensor strain error is expected to behave differently; it is hypothesized that this type of error will increase towards a maximum asymptote as the number of strain sensors is increased. This is expected based on the following reasoning: when the number of strain sensors is smaller than or equal to the number of basic load cases, the CM method can, in theory, determine a set of basic load case coefficients that will result in a perfect match between the strains of the reconstructed deformation and the strains that were determined by the sensors. As the number of sensors increases beyond the number of basic load cases, this is no longer possible, and the method will instead have to determine the best possible and therefore imperfect match.

### 2.8. Computational Efficiency

A shape sensing algorithm should ideally run in real-time to reconstruct displacements, strains, stresses, and loads as quickly as measurement data is obtained and to provide live feedback about the health of the structure. To this end, the computational efficiency of this implementation of the CM and iFEM algorithms were evaluated by running the algorithm for blades of different sizes: the blade length was varied, leading to a corresponding change in the number of elements and nodes in the mesh, the number of basic loads, and the number of sensors, while maintaining the same base section, taper and twist distribution along the length of the blade, and the NACA 6412 thickness profile.

For the numerical work presented in this paper, all the steps of the CM algorithm shown in Figure 4 take time to execute. However, for real applications, the steps before “Solve for coefficients & reconstruct loads” are irrelevant for real-time performance because they are carried out only once. Of the remaining steps, solving for coefficients is the most significant: the S matrix is dense, which makes the $S^{T}S$ matrix from Equation (6) dense as well. $S^{T}S$ is an m-by-m square matrix, where m is equal to the number of basic load cases. Solving a dense matrix requires order $O\left(m^{3}\right)$ floating point operations (flops): as the number of basic load cases doubles, the solution time is expected to octuple.

Similarly, all the steps of the iFEM algorithm shown in Figure 6 take time to execute as well, but all the steps before “Solve for estimate of iFEM displacements” are irrelevant for the real-time performance because they are also carried out only once. While the A matrix from Equation (13) is sparse, solving it is still the most significant remaining step. A is an 3n-by-3n square matrix, where n is the number of nodes in the mesh and has a bandwidth of k. Solving this sparse matrix requires order $O(n*k^{2})$ flops: as the number of nodes doubles, the solution time is expected to quadruple.

However, some optimizations can be performed for both algorithms. For the CM algorithm, repeating the same solution for new measurements requires only order $O\left(m^{2}\right)$ flops, so the solution time should theoretically only quadruple as the number of basic load cases doubles. A similar optimization exists for the iFEM algorithm: repeating the same solution for a matrix with a known bandwidth requires only order $O\left(n*k\right)$ flops.

The facts that different shape sensing problems will have different values for n and k and that an arbitrary value can be chosen for m make it hard to compare the relative performance of the CM and iFEM algorithms a priori. To provide an indication of the performance of both algorithms, they were evaluated numerically using the same twisted blade that was used for the convergence study, scaled along its length to vary the number of mesh nodes. The number of strain sensors and basic load cases were adjusted proportionally to the number of mesh nodes.

It is expected that the computational time will grow linearly for the iFEM algorithm as n increases while k stays constant. The first solution time for the CM algorithm is expected to increase cubically, and the repeat solution time is expected to increase quadratically. As m is much smaller than n, the solution times for CM are expected to initially be lower than those for iFEM, but they will eventually exceed those for iFEM as n and m increase.

The evaluation of the relative computational efficiency of the iFEM and CM algorithms in this paper is intentionally biased towards the iFEM algorithm: the number of basic load cases m does not actually need to grow at all or as fast as the number of nodes n, as the structural mesh is made finer and more detailed. Additionally, the number of nodes through the width and thickness of the blade was kept constant to keep the bandwidth k constant.

## 3. Results

CM and iFEM analyses were carried out for all 22 load cases listed in Table 1. Note that some load cases use the same structure with a different mesh, applied load, or Poisson’s ratio. Each structure was instrumented with a distribution of displacement and strain sensors, as shown in Figure 1.

FEM analyses were carried out for all 22 benchmark load cases as well. These are needed to determine the sensor data that would be measured during the actual loads and, for comparison to the reconstructed deformation, to determine their accuracy. To show that the results of these FEM analyses are accurate, key displacements from the results are compared to their analytical counterparts from MacNeal and Harder [55] in Figure 7. The patch tests are not included in this table because their displacements were prescribed:

These results closely match those of MacNeal and Harder themselves for the 20-node hexahedral elements with reduced integration, with most errors below 10%, corresponding to an A or B grade for this element type on the grading scale of MacNeal and Harder.

To obtain each set of CM results, FEM analyses were carried out for all the basic load cases for each structure, which were used to simulate what the sensor data would be for each of these basic load cases. An example FEM result is shown in Figure 8, where the value of the E11 strain component is plotted on the surface of the twisted beam for the same basic load case that was shown in Figure 5:

To show the correspondence between the FEM results and the simulated sensors more clearly, using a larger number of sensors, these results were plotted together in Figure 9 for one of the basic load cases applied to the blade:

The reconstructed forces for each actual load case were evaluated using FEM; Figure 10 shows an example of such a reconstruction: the tractions for this load case should be a uniform distribution over the bottom surface of the blade, as shown in the top part of the figure. The reconstructed tractions shown in the bottom left are not a perfect match but appear to be a good approximation, achieving a distribution that is close to uniform and having a magnitude that seems to mostly be within 10% of the expected values. To evaluate this more precisely, the relative traction errors are shown in the bottom right of the figure. These errors compare the expected and reconstructed values for each individual traction. This plot closely matches the visual impression of the results: the largest error was found to be 12.7% in a small region towards the tip of the blade, with the rest of the blade achieving an error of less than 10%.

### 3.1. Convergence Study

A convergence study was performed to determine whether the meshes have sufficient quality to produce reliable results. The results indicated a mesh quality like what was achieved by Slater et al. [62]. The FEM and CM values were generally very close matches, as can be seen in Figure 11, which was expected from the results for the finest grid spacing.

All the error bands were well below 10%, so they are considered to be acceptable. The error bands for the total force can even be considered good, as they were below 1%. These error bands indicate that the peak displacements are reasonably accurate approximations of the values that would be obtained with an infinitely fine mesh and that the total forces are generally very accurate approximations.

### 3.2. Sufficient Number of Strain Sensors

Several CM analyses were carried out using the representative aerospace structure to provide an indication of the number of strain sensors that were needed to obtain good results. As the definition of good depends on the application, this can only be an indication. In Figure 12 below, the force errors were plotted against the number of strain sensors that were used for the CM analysis for each reconstructed load case in the X direction.

The force errors in the other directions, the coefficient errors, the strain errors, and the displacement errors followed a very similar and expected pattern: each error generally decreased as the number of strain sensors increased. The error magnitudes decreased from unacceptable levels at 30 sensors down to 16% (Load 5), or around 2% at 100 sensors. 16% error was not considered acceptable, while 2% was acceptable. The error magnitudes stabilized around 11% (Load 5) and 1.9% at 300 strain sensors, which would be an expensive number of sensors when metallic foil strain gauges are used, but very feasible for a single fiber optic strain sensor placed along the surface of a structure.

Only the strain error for the sensors showed a different pattern. As expected, and as shown on the right in Figure 13, these errors increased as the number of sensors increased, leveling off to a stable magnitude. Note again the difference in scale, compared to the previous figures:

The hypothesis about the behavior of the sensor strain errors was correct: it increases as the number of strain sensors increases. These errors are all considered to be good, as they are below 1% for any number of sensors, increasing from 0.3% or lower at 30 sensors, to 0.9% or lower at 100 sensors and beyond. They also approach the maximum value as expected. For 300 sensors, the error was found to be 0.36% on average and 0.88% in the worst case, which are still considered to be good values.

While precise criteria for the magnitude of the errors would depend on the specific application, the errors that were achieved here seem acceptable: for 300 strain sensors, which is a number of data points that is practical for real applications [3,33,34,46,47,48,49,63,64,65], the average errors were 4.7% for the forces, 2.0% for the coefficients, 0.036% for the displacements, and 0.32% for the strains. For specific applications, these errors could be improved further by increasing the number of sensors, using a finer mesh, using more basic load cases, and possibly by optimizing the placement of the sensors.

### 3.3. Errors between Reconstructed and Actual Load Cases

The results that were reconstructed with the CM algorithm described above can be compared to the FEM results for each actual load case using Equation (9) to produce the $\varepsilon_{coefficient}$ error. Similarly, the $\varepsilon_{force}$, $\varepsilon_{displacement}$, $\varepsilon_{strain}$, and $\varepsilon_{sensor\;strain}$ errors can be calculated. The resulting values are presented in Figure 14 for each actual load case, with color coding to indicate the quality.

Nearly all the results were found to be acceptable: most errors fell below 1%, and nearly all fell below 10%. Only two load cases had force errors above 10%: the cantilever beam made of regular elements and loaded in out-of-plane shear had a force error of 14.47%, and the curved beam loaded in out-of-plane shear had a force error of 48.35%.

These results may seem inconsistent at first, as the force and coefficients are sometimes significantly higher than the displacement, strain, and sensor strain errors. The reason for this becomes apparent when the results with the greatest errors are examined in more detail. For example, Figure 15 and Figure 16 show the curved beam that is loaded in out-of-plane shear, which is the load case with the greatest apparent inconsistency between the force and coefficient errors.

The expected loads for this load case are zero everywhere on the structure except for a unit load in the z-direction on the tip, as shown in red in Figure 15. While the reconstructed loads in the z-direction are a close match for the actual loads that were used for the FEM results, there are additional reconstructed loads in the positive and negative x and y directions, as shown in Figure 16. These unexpected nonzero loads approximately cancel out, resulting in little effect on the displacement and strain errors, but they do have a significant effect on the force and coefficient errors.

To compare the accuracy of the CM and iFEM algorithms the relative errors for each load case resulting from each algorithm can be compared, which is performed in Figure 17 below. The force and coefficient errors have been excluded here because these do not exist for the iFEM algorithm. The strain sensor errors have also been excluded because they are nearly identical to the strain errors in all cases. Averaged over all the load cases examined here, the CM algorithm achieved average displacement and strain errors of less than 0.01%, whereas the iFEM algorithm had an average displacement error of 21% and an average strain error of 99%.

### 3.4. Computational Efficiency

Figure 18 below shows the computational efficiency for the CM and iFEM algorithms by plotting the computational time against the number of strain sensors. The length of the blade, the number of nodes along the length, and the number of basic load cases were adjusted in proportion to the number of strain sensors to maintain the ratio between the number of sensors and the number of basic load cases and to maintain the same fine mesh.

The iFEM and CM analyses were run consecutively to ensure that there was no influence on the runtime between the two methods. All the analyses were carried out on the same laptop to ensure a fair comparison. A 2020 HP Spectre x360 Convertible 15-eb0100nd with 16.0 GB of DDR4 RAM and an Intel Core i7-10510U 2.30 GHz 4 core CPU.

The data sets were fitted to power curves, resulting in the equations $y=1.67\cdot 10^{-9}\;x^{2.84}$ for the solution time of the CM algorithm using the MKL solver, $y=3.05\cdot 10^{-10}\;x^{3.04}$ for the first solution time of the CM algorithm using the in-house solver, and $y=1.19\cdot 10^{-10}\;x^{2.39}$ for the second solution time of the CM algorithm, where y is the time in seconds and x is the number of basic load cases. The fit for the iFEM algorithm was $y=8.08\cdot 10^{-4}\;x^{1.01}$. These curves are close to the cubic, quadratic, and linear growth rates that were expected.

The adjusted coefficients of determination, $R_{adjusted}^{2}$, were 0.994, 0.998, 0.94, and 0.92, which indicates that the fitted curves can account for 99.4%, 99.8%, 94%, and 92% of the observed variation in the solution time data. For the purposes of giving an indication of the relative performance of the two algorithms, this is considered satisfactory.

Even with the handicaps imposed on the implementation of the CM algorithm, i.e., growing the number of basic load cases as fast as the number of nodes and keeping the number of nodes through the width and thickness of the blade constant to keep the bandwidth k constant, it still has a computational performance similar to the iFEM algorithm for this representative aerospace structure and outperforms the iFEM algorithm by a factor of approximately 100 when using the repeat solution.

## 4. Discussion

One of the primary takeaways from this study is that the forces, displacements, and strains (for both the Gauss points and the sensors) that were reconstructed by the calibration matrix method closely match the actual values for the expected load cases, with a number of sensors that would be feasible to apply to a real structure. New sensor data can also be evaluated in a fraction of a second, even when thousands of basic load cases and sensors are used. These facts, taken together, indicate that the CM algorithm would be a good method for real-time reconstruction of the applied forces and the corresponding displacement and strain distributions.

The errors are larger for slender structures that are loaded by out-of-plane shear loads due to the reconstruction of spurious forces. While these differences are small enough to not pose a problem for the applicability of the method, this behavior suggests that the methodology could benefit from further improvements to handle these and similar load cases, perhaps by distributing the sensors strategically, refining the mesh, or refining the number of basic load cases.

The CM and iFEM analyses can use any number of strain sensors, which can be mounted on the surface and/or distributed throughout the volume of the structure. As expected, the use of a greater number of sensors resulted in smaller errors for the reconstructed forces, coefficients, displacements, and strains when all other aspects of the analysis were kept the same.

For the CM algorithm to be of practical use, it should be able to achieve an acceptable error with a number of sensor data points that would be realistic to implement in a real structure, provided that a sufficiently good mesh (as determined by a convergence study) and enough basic load cases are used as well. Acceptable error values will depend on the application; here, below 10% was considered acceptable and below 1% was considered good. The number of sensors that would be realistic will also depend on the application. This algorithm was intended for use in combination with modern fiber optic strain sensors (FOSS), which can measure unidirectional strain at thousands of points with a single optical fiber [63]. As the algorithm uses simulated strain sensors that measure all six components of strain, it is realistic to measure strain at hundreds of points.

With a realistic number of strain sensors, the CM algorithm was able to achieve a good level of accuracy for the reconstructed displacements, strains, and strain sensors for all the benchmark problems, a good or acceptable level of accuracy for the reconstructed coefficients for all the benchmark problems, and a good or acceptable level of accuracy for the reconstructed forces for most of the benchmark problems. Levels of accuracy that were not considered acceptable were only obtained for the reconstructed forces for slender structures loaded in out-of-plane shear.

The errors for the strain sensor values were expected to become slightly worse as a larger number of sensors were used because the CM algorithm tries to match the actual strain sensor values with a linear combination of basic strain sensor values. The degrees of freedom of this linear combination are the basic load case coefficients. As the number of sensors increases, the number of these coefficients stays the same, so the number of degrees of freedom stays the same. When the same number of degrees of freedom are used to reconstruct a larger number of sensor values, the quality cannot go up. Still, the quality was expected to level out, which it did because the sensor values are not random: adjacent sensors will have similar values.

The computational efficiency results were somewhat surprising: while the fitted orders of time complexity were close to the expected values for the iFEM solution ($O(x^{1.01})$ instead of $O\left(x^{1}\right))$ and the first CM solution ($O(x^{3.04})$ instead of $O\left(x^{3}\right))$, it was higher than expected for the repeated CM solution, at $O(x^{2.39})$ instead of $O\left(x^{2}\right)$. The adjusted coefficient of determination indicated a good fit for the curve, suggesting that this is a real effect that could be investigated and perhaps improved further. The CM solution using the MKL solver, while slower in absolute terms, had a better than expected time complexity ($O(x^{2.84})$ instead of $O\left(x^{3}\right))$. This suggests that further optimizations are possible, which would be beneficial for problems that are larger than the ones studied here.

A potential explanation for the larger than expected time complexity for the repeated CM solution could be that it’s due to caching: as the amount of data that is processed increases, it may no longer fit in the CPU cache. Instead, it needs to be stored in and retrieved from RAM, which slows down the solution process as the number of basic load cases increases. This would also occur for the first CM and iFEM solution processes, but because the repeat solution takes a smaller amount of time, such caching delays could become significant.

Still, real-time performance has been demonstrated: the repeat solution for new sets of input data requires less time than the initial solution, by a factor of several hundred (depending on the problem size). Even for numbers of load cases that are larger than what is required to reach acceptable levels of quality, the repeat solution is completed in a fraction of a second.

## 5. Conclusions

It has been demonstrated that the CM algorithm can reconstruct load cases more accurately than the iFEM algorithm for various benchmark problems. It is possible to obtain good results with the CM method even when only a small number of basic load cases and sensors are used, provided that certain conditions are met: the number of sensor data points should meet or exceed the number of basic load cases, and the collection of basic load cases should be broad enough to allow a good approximation to be made by linear combinations of these basic load cases for each actual load case that may occur.

The CM method is also fast, as most of the computation can be performed in advance. After carrying out the initial effort of running FEM analyses for each basic load case, simulating the resulting sensor data, and collecting this data in the S matrix, only the calculation of the coefficients needs to be repeated for a new set of sensor data. The pre-computed solution for a relatively small system of linear equations can be re-applied to each new set of sensor data from actual load cases in a fraction of a second.

The main recommendation for future research is to test the CM method with more scenarios, both numerically and experimentally, to further verify and validate its performance, including a validation of the real-time performance by streaming in real sensor data to the algorithm and streaming out the reconstructed load, displacement, and strain distributions to visualization software and other systems.

## Figures and Tables

**Figure 1 sensors-24-03562-f001:**
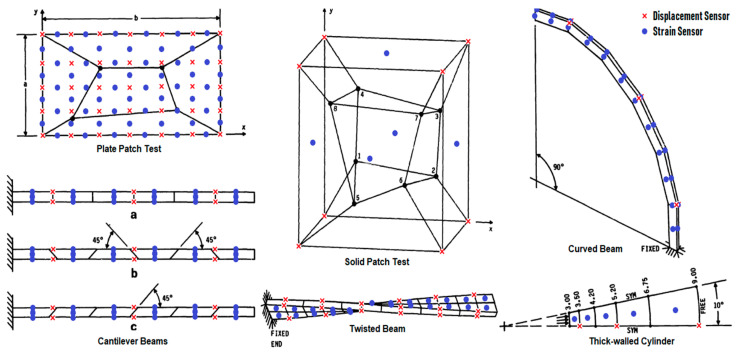
Benchmarks problem from de Mooij et al. [37], based on a selection from MacNeal and Harder [16], augmented with sensors: displacement sensors are in red, strain sensors are in blue. Cantilever beams a-c are composed of regular, trapezoid and parallelogram elements, respectively.

**Figure 2 sensors-24-03562-f002:**
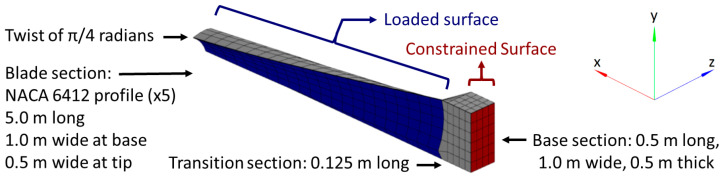
Isometric view of twisted and tapered blade with NACA 6412 cross-sectional profile, divided into 648 hexahedral elements. Fully constrained areas are highlighted in red and loaded areas are highlighted in blue.

**Figure 3 sensors-24-03562-f003:**
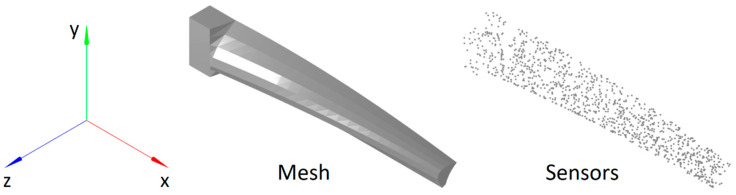
A 3D illustration of the mesh and the simulated strain sensors.

**Figure 4 sensors-24-03562-f004:**
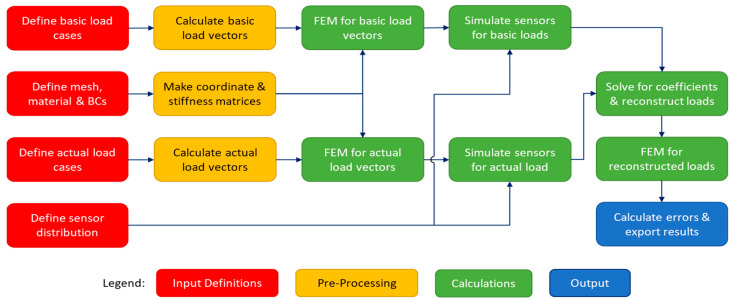
CM algorithm flowchart for numerical approach.

**Figure 5 sensors-24-03562-f005:**
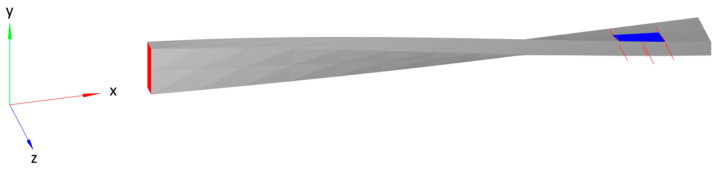
Example of a basic load case: a unit force along the z-axis, applied to a region on the top surface of the twisted beam, marked in blue. The boundary condition, the fully constrained root of the beam, is marked in red.

**Figure 6 sensors-24-03562-f006:**
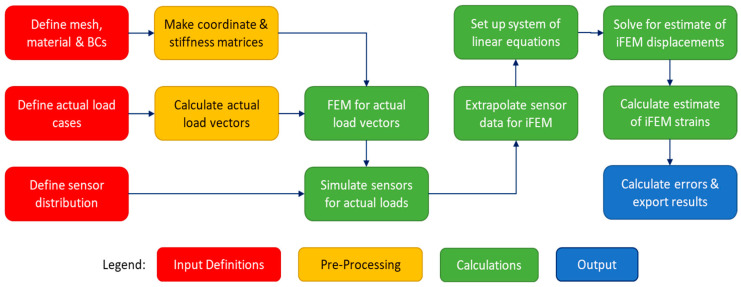
iFEM algorithm flowchart for numerical approach.

**Figure 7 sensors-24-03562-f007:**
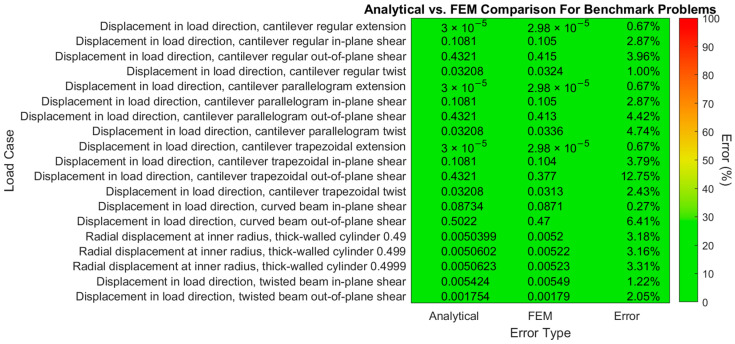
Comparison of analytical and FEM results for the MacNeal and Harder [55] benchmark problems.

**Figure 8 sensors-24-03562-f008:**
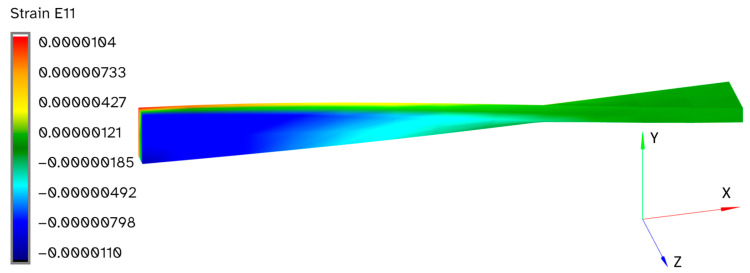
Example FEM result for the basic load case shown in Figure 5. The value of the E11 strain component is plotted on the surface of the twisted beam.

**Figure 9 sensors-24-03562-f009:**
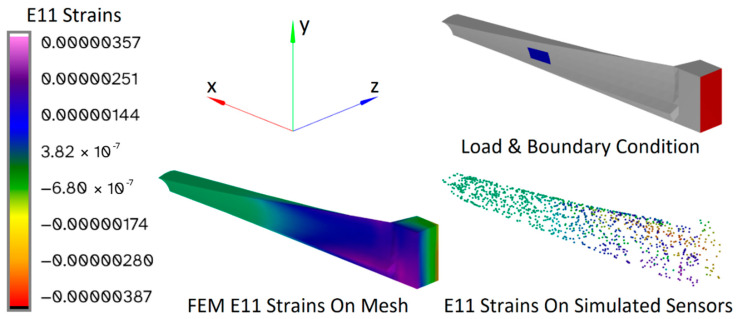
Example FEM result for basic load case 60 a unit force in the z-direction, applied with surface tractions. Shown are the loaded and constrained regions, the E_11_ strains on the mesh, and the E_11_ strains of the simulated sensors.

**Figure 10 sensors-24-03562-f010:**
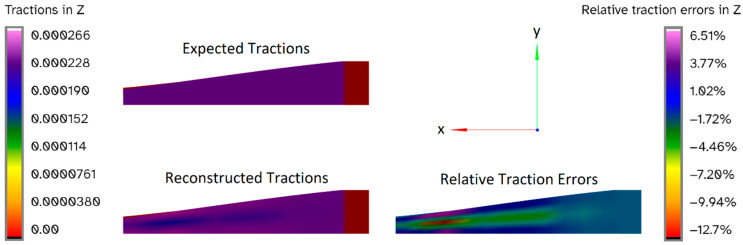
Expected load case 1z, uniform load: expected tractions, reconstructed tractions, and relative traction errors.

**Figure 11 sensors-24-03562-f011:**
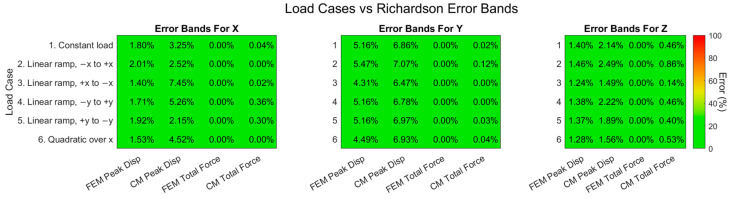
Load Cases vs. Richardson Error Bands.

**Figure 12 sensors-24-03562-f012:**
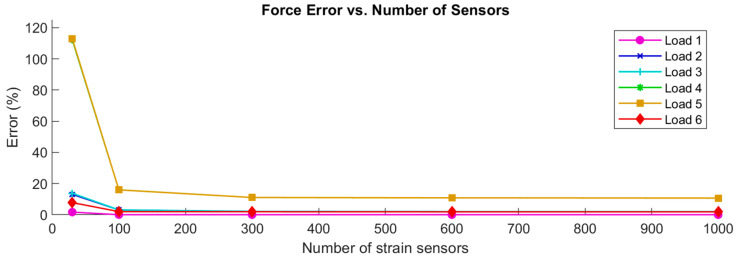
Force errors vs. number of sensors for the six reconstructed load cases in the X direction.

**Figure 13 sensors-24-03562-f013:**
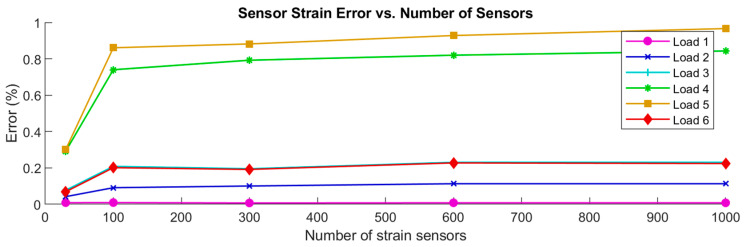
Sensor strain errors vs. number of strain sensors for 6 reconstructed load cases in the X direction.

**Figure 14 sensors-24-03562-f014:**
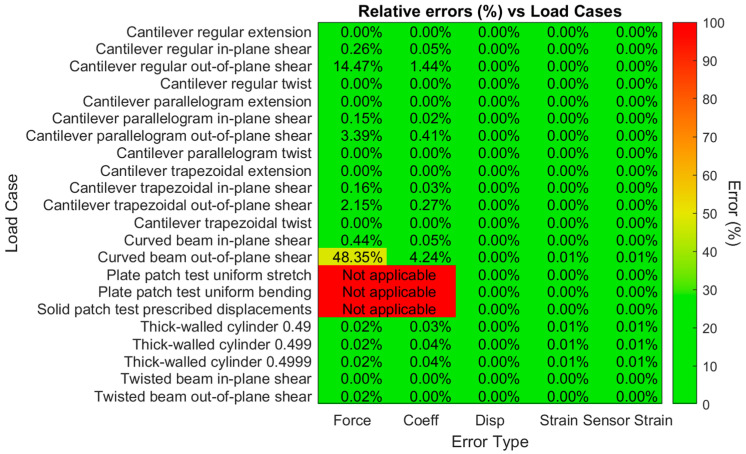
Calculated errors between reconstructed and actual results for forces, coefficients, displacements, strains, and sensor strains per actual benchmark problem load case for the Calibration Matrix method.

**Figure 15 sensors-24-03562-f015:**
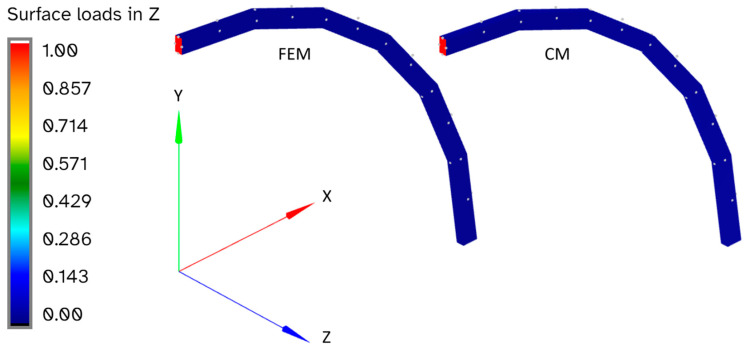
Comparison of surface loads (N) for a curved beam loaded in out-of-plane shear.

**Figure 16 sensors-24-03562-f016:**
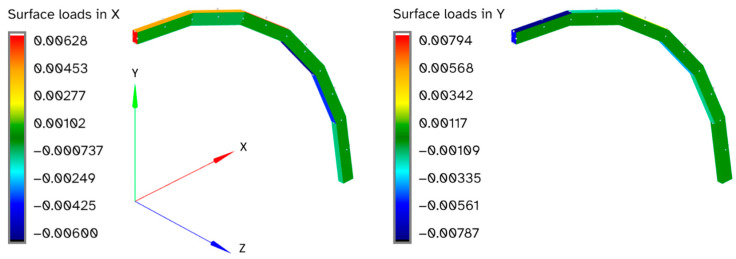
Reconstructed surface loads (N) in x and y-directions for a curved beam loaded in out-of-plane shear.

**Figure 17 sensors-24-03562-f017:**
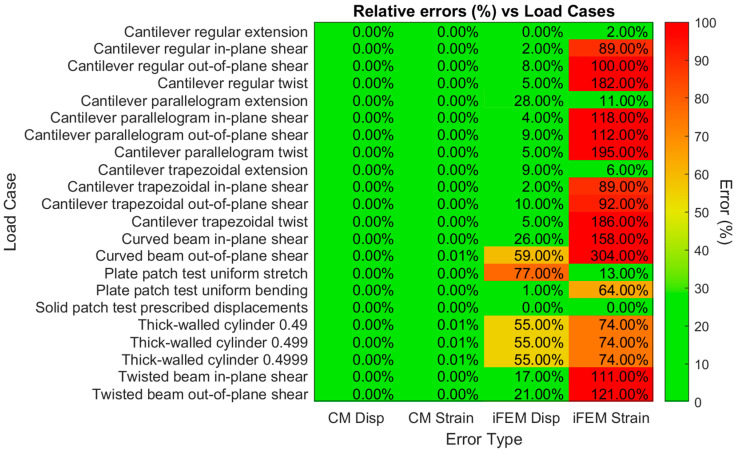
Comparison of relative errors of the displacements and strains for the CM and iFEM reconstructions of each benchmark problem.

**Figure 18 sensors-24-03562-f018:**
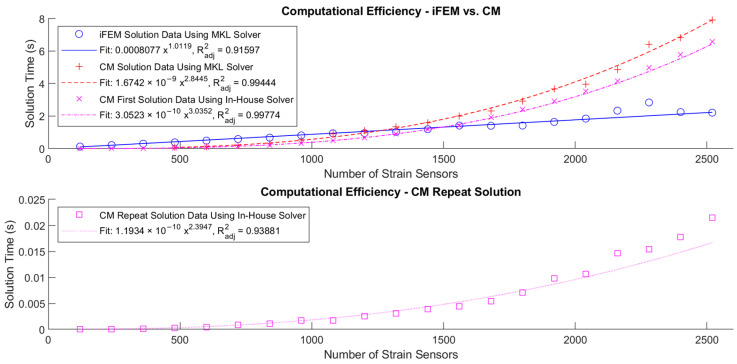
Computational efficiency: solution time vs. number of strain sensors (which was kept equal to the number of basic load cases here) for the first solution of CM and iFEM and the repeat solution of CM.

**Table 1 sensors-24-03562-t001:** Benchmark problem properties from de Mooij et al. [37], adapted from MacNeal and Harder [55].

Name	Element Shape	E (Pa) and (ν) (-)	Dimensions (m)	Load Case and BCs	Mesh Size
Cantilever	Regular, Parallelogram, Trapezoidal	1.0 × 10^7^ (0.30)	6.0 × 0.2 × 0.1	Unit loads (N) on tip: extension, in-plane shear, out-of-plane shear or twist (unit moment (Nm)) and clamped	6 elements
Curved Beam	Regular	1.0 × 10^7^ (0.25)	radius: 4.12 to 4.32 thickness: 0.1 span: 90°	Unit loads (N) on tip: in-plane shear or out-of-plane shear and clamped	6 elements
Plate Patch Test	Irregular	1.0 × 10^6^ (0.25)	0.12 × 0.24 × 0.001	Prescribed uniform stretch (m) or bending (m)	5 elements
Solid Patch Test	Irregular	1.0 × 10^6^ (0.25)	1 × 1 × 1	Prescribed boundary displacements (m)	7 elements
Thick-walled Cylinder	Regular	1000 (ν: 0.49, 0.499, 0.4999)	radius: 3.0 to 9.0 thickness: 1.0	Unit pressure (Pa) at the inner radius	5 elements per 10°
Twisted Beam	Regular	29.0 × 10^6^ (0.22)	12.0 × 1.1 × 0.32 twist: 90°	In-plane shear (N) or out-of-plane shear (N) and clamped	12 × 2

**Table 2 sensors-24-03562-t002:** Convergence study parameters: number of elements along each axis and in total for three levels of mesh refinement.

Normalized Grid Spacing	Elements along x Axis	Elements along y Axis	Elements along z Axis	Elements
1	80	16	8	10,240
2	40	8	4	1280
4	20	4	2	160

**Table 3 sensors-24-03562-t003:** Load distributions for the representative aerospace structure: numbers, descriptions, and equations.

**Load Case**	Description	Load Magnitude Distribution (N)
1x, 1y, 1z	Constant load	0.001
2x, 2y, 2z	Linear ramp, −x to +x	$0.002*(x-L_{base})/L_{blade}$
3x, 3y, 3z	Linear ramp, +x to −x	$0.003*\left(1-(x-L_{base})/L_{blade}\right)$
4x, 4y, 4z	Linear ramp, −y to +y	$0.004*y/W_{blade}$
5x, 5y, 5z	Linear ramp, +y to −y	$0.005*\left(1-y/W_{blade}\right)$
6x, 6y, 6z	Quadratic over x	$0.006*\left(1-{\left((x-L_{base})/L_{blade}\right)}^{2}\right)$

## Data Availability

The original contributions presented in the study are included in the article, further inquiries can be directed to the corresponding author. The C++ implementations of the Calibration Matrix algorithm and the iFEM algorithm presented herein are under embargo until the graduation of Cornelis de Mooij.

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
