# Peer review of "A Critical Comparison of Shape Sensing Algorithms: The Calibration Matrix Method versus iFEM"

_sensors, 2024, doi:10.3390/s24113562_

Round 1

Reviewer 1 Report

Comments and Suggestions for Authors

It's quite an interesting research; I just suggest adding data in the abstract to explain why the calibration matrix is more accurate than iFEM.

Adjust the formatting to avoid words being cut off by lines

Reviewer 2 Report

Comments and Suggestions for Authors

Comment 1: In Table 1, what’s the unit for dimensions? Meter?

Comment 2: For Figures 12 and 13, what’s the Error (%) for 100 strain sensors? Is it an acceptable values? The authors can expand discussion here.

Comment 3: This article compared two shape sensing algorithms and concluded Calibration Matrix (CM) method could provide more accurate results than the iFEM algorithm. The methods were clearly discussed in the article and the findings are interesting.

Comments on the Quality of English Language

The language is good, not major issue found

Reviewer 3 Report

Comments and Suggestions for Authors

The authors propose a modified version of the CM method, where instead of directly solving for numerous individual nodal pressures, the approach tackles a relatively smaller set of basic load case coefficients. Another innovation lies in employing Ordinary Least Squares (OLS) rather than the pseudo-inverse. However, the paper lacks significant novelty beyond refining an existing approach. Validation relies solely on simulation data, and the algorithm's requirement for hundreds of sensor data points poses a practical challenge. Although the authors acknowledge this limitation, neither the methodology nor the core concept undergoes validation via experimental setup. Additional points for consideration include:

·      The CM algorithm assumes the actual load case to be a linear combination of various basic load cases. However, it's unclear how this assumption holds in cases of non-linearity.

·      The differentiation degree among various strain sensors raises questions, particularly regarding the potential redundancy of information from closely located strain points, which could lead to dependence among matrix S rows.

·      Clarification is needed on the meaning of "c-optimal" in Equation (7), with an expectation for the denominator to reference c_actual.

·      Equations lack clarity without accompanying dimensions. It's advisable for the authors to include descriptions/dimensions of each vector/matrix following their respective equations to enhance comprehension.

Based on the overall significance, novelty, and practicality, I do not recommend this paper for publication.
